# Rapid assembly of SARS-CoV-2 genomes reveals attenuation of the Omicron BA.1 variant through NSP6

Taha Y. Taha [1,10] ✉, Irene P. Chen [1,2,10], Jennifer M. Hayashi [1,10], Takako Tabata[1,10], Keith Walcott [1], Gabriella R. Kimmerly[1], Abdullah M. Syed[1,3], Alison Ciling [1,3], Rahul K. Suryawanshi[1], Hannah S. Martin[1,4], Bryan H. Bach [3], Chia-Lin Tsou[1], Mauricio Montano[1], Mir M. Khalid[1], Bharath K. Sreekumar[1], G. Renuka Kumar[1], Stacia Wyman [3], Jennifer A. Doudna [1,3,4,5,6,7,8] & Melanie Ott [1,2,9] ✉

Although the SARS-CoV-2 Omicron variant (BA.1) spread rapidly across the world and effectively evaded immune responses, its viral fitness in cell and animal models was reduced. The precise nature of this attenuation remains unknown as generating replication-competent viral genomes is challenging because of the length of the viral genome (~30 kb). Here, we present a plasmid-based viral genome assembly and rescue strategy (pGLUE) that constructs complete infectious viruses or noninfectious subgenomic replicons in a single ligation reaction with >80% efficiency. Fully sequenced replicons and infectious viral stocks can be generated in 1 and 3 weeks, respectively. By testing a series of naturally occurring viruses as well as Delta-Omicron chimeric replicons, we show that Omicron nonstructural protein 6 harbors critical attenuating mutations, which dampen viral RNA replication and reduce lipid droplet consumption. Thus, pGLUE overcomes remaining barriers to broadly study SARS-CoV-2 replication and reveals deficits in nonstructural protein function underlying Omicron attenuation.

Severe acute respiratory syndrome coronavirus 2 (SARS-CoV-2) is the causative agent of the coronavirus disease 2019 (COVID-19) pandemic. The pandemic continues as a major public health issue worldwide. As of October 2022, more than 600 million people have been infected with it and more than 6.5 million have died[1]. The continuous emergence of viral variants represents a major threat to our pandemic countermeasures due to enhanced transmission[2–4] and antibody neutralization escape[5].

The emergence of the Omicron variant (BA.1) in November 2021 was especially concerning due to the large number of mutations throughout the genome (53 nonsynonymous mutations) and 34 mutations in the Spike protein alone. While Omicron infections spread significantly more rapidly than previous variants, they are associated with fewer symptoms and lower hospitalization rates[6–8]. Accordingly, the Omicron variant is attenuated in cell culture[9–12] and animal models of infection[13–15]. An evolutionary tradeoff appears to exist between

[1]Gladstone Institutes, San Francisco, CA, USA. [2]Department of Medicine, University of California, San Francisco, CA, USA. [3]Innovative Genomics Institute, University of California, Berkeley, CA, USA. [4]Department of Chemistry, University of California, Berkeley, CA, USA. [5]Department of Molecular and Cell Biology, University of California, Berkeley, CA, USA. [6]Howard Hughes Medical Institute, University of California, Berkeley, CA, USA. [7]Molecular Biophysics and Integrated Bioimaging Division, Lawrence Berkeley National Laboratory, Berkeley, CA, USA. [8]California Institute for Quantitative Biosciences (QB3), University of California, Berkeley, CA, USA. [9]Chan Zuckerberg Biohub – San Francisco, San Francisco, CA, USA. [10]These authors contributed equally: Taha Y. Taha, Irene P. Chen, Jennifer M. Hayashi, Takako Tabata. ✉e-mail: taha.taha@gladstone.ucsf.edu; melanie.ott@gladstone.ucsf.edu

increased viral spread and diminished infection severity in the context of an increasingly immunized human population. This tradeoff may have arisen only recently as adaptive evolution of SARS-CoV-2 prior to the emergence of Omicron was mainly characterized by purifying selection[16].

SARS-CoV-2 is an enveloped positive-strand RNA virus in the family *Coronaviridae* in the order *Nidovirales*[17]. Its ~30 kb genome contains at least 14 known open reading frames (Fig. 1A). The 5' two-thirds of the genome encompass ORF1a and ORF1ab that code for polyprotein 1a and 1ab, respectively, which are subsequently proteolytically processed to 16 non-structural proteins (NSPs) by the two virally encoded proteases (NSP3 and NSP5) and execute replication and transcription of the viral genome (reviewed in[18]). The 3' one-third of the genome include the viral structural and accessory proteins. SARS-CoV-2 particles are composed of four structural proteins including Spike (S), Envelope (E), Membrane (M), and Nucleocapsid (N)[19–21]. The S protein mediates viral entry and fusion by binding the ACE2 receptor on cells and is the subject of evolutionary selection to evade neutralization by vaccine- and infection-elicited antibodies[5]. The viral accessory proteins (ORF3a, 3b, 6, 7a, 7b, 8, 9b, 9c, and 10) have diverse functions contributing to infectivity, replication, and pathogenesis and other unknown functions (reviewed in[22]).

To study SARS-CoV-2 attenuation and the full range of mutations along the Omicron genome, it is necessary to construct full-length recombinant viruses or near full-length replicons. Replicons lack critical structural proteins such as Spike and cannot spread in cultures due to missing infectious particle production[23,24]. They can, however, autonomously replicate viral RNA, either after straight-forward transfection of the replicon genomes or after single round infections with viral particles generated with transiently provided structural proteins[23].

Constructing SARS-CoV-2 recombinant viruses or replicons in a timely manner is challenging due to the length of the viral genome (~30 kb) and the presence of several toxic viral sequences[25] that limit standard molecular cloning strategies. The key hurdle is the faithful and timely assembly of the complete viral genome from multiple subgenomic fragments. Several approaches have been reported to assemble SARS-CoV-2 infectious clones, each having contributed important insight into the biology of SARS-CoV-2 (reviewed in[26]). These involve either ligation- or PCR-based approaches and include the synthetic circular polymerase extension reaction (CPER) approach[27,28], the ligation of synthetic fragments using unique restriction enzymes in the SARS-CoV-2 genome[23,29,30], and ligation of synthetic or cloned fragments using type IIs restriction enzymes[25,31,32].

The CPER assembly approach, adapted from tickborne encephalitis virus research[33] and widely used in viral reverse genetics, is fast when a suitable template containing desired mutations for amplification is available but has limited capacity to simultaneously introduce a large number of new mutations directly from cDNA templates as each mutation involves a separately amplified fragment. Recently, Kim et al. devised an optimized CPER approach where multiple new mutations can be introduced simultaneously in multiple fragments[34]. Utilization of restriction sites for in vitro ligation of subgenomic fragments into a linear cDNA or plasmid was first described for brome mosaic virus[35] and has been widely used to generate full-length coronaviral genomes such as mouse hepatitis virus[36]. It is a straight-forward molecular cloning technique but involves step-wise incubation and purification steps and often results in low yields of the full-length ligated genome. This method also precludes rational fragment design as the location of the restriction sites dictates the fragment borders. There remains a need for a rapid, reliable and rationally designed cloning strategy to make SARS-CoV-2 reverse genetic applications widely available and enable timely characterization of emerging SARS-CoV-2 variants.

To overcome these limitations, we developed plasmid-based viral genome assembly and rescue (pGLUE), a method that takes advantage of the Golden Gate assembly method to seamlessly digest and ligate viral sequences in a single-pot reaction. Golden Gate uses type IIs restriction enzymes that cleave outside their recognition sequences and combines ligation and digestion with temperature cycling to carry out reliable and rapid assembly of multiple fragments in a few hours[37,38].

Using pGLUE and an optimized virus rescue protocol, we de novo constructed several naturally occurring Delta-Omicron chimeric infectious clones and found that both, mutations in ORF1ab and Spike, contribute to Omicron attenuation. To precisely map which mutations attenuate RNA replication, a large series of chimeric replicons was generated that lacked Spike. These revealed that attenuated RNA replication in Omicron mapped to mutations in NSP6, which caused diminished lipid droplet consumption otherwise fueling viral RNA replication. Thus, access to rapidly generated replicating SARS-CoV-2 genomes provided important new insight into SARS-CoV-2 biology.

## Results

### Golden gate assembly enables rapid cloning of SARS-CoV-2 variants

To determine which parts of the Omicron genome contribute to the attenuated phenotype, we designed and developed pGLUE (plasmid-based viral genome assembly and rescue): a rapid method to generate SARS-CoV-2 molecular clones with Golden Gate assembly (Fig. 1A). The SARS-CoV-2 genome was newly divided into 10 fragments to enable quick and reliable cloning of mutations. The fragments were rationally designed to each encompass distinct SARS-CoV-2 proteins and ORFs, which facilitates the interrogation of mutations in individual viral proteins and the construction of chimeric viruses and replicons (Supplementary Fig. 1). All fragments were stable in bacteria, grew to high copy numbers, and were amenable to standard molecular cloning approaches. Typically, mutagenesis of these fragments took no longer than 4 days on average (including primer synthesis, PCR, assembly, transformation, plasmid prep, and sequencing) by utilizing an optimized Gibson assembly mutagenesis method[39]. In addition, to ensure lack of undesirable mutations, all plasmids were nanopore sequenced within ~20 h with at least >x250 coverage (Supplementary Fig. 2a). The fragments were assembled along with a bacterial artificial chromosome (BAC) vector to enable growth of toxic sequences within the SARS-CoV-2 genome in bacteria, such as those found in the second, third, and seventh fragment of a previously reported reverse genetics system[25,31,32]. At the 5' end, the vector carries T7 and CMV promoters with the T7 promoter nested in between the TATA box sequence of the CMV promoter and the SARS-CoV-2 RNA transcription start site, which is located at position +27 downstream of the TATA box. This enables DNA- or RNA-based launches of viral production. The 3' end of the vector contains a hepatitis delta virus ribozyme (HDVrz) and simian virus 40 (SV40) polyA sequences for efficient 3' RNA processing.

The Golden Gate assembly reaction was highly efficient in generating the assembled genome and within 30 cycles (~5–6 h) shifted almost the entire DNA content into the slower migrating assembly product (Fig. 1B). Sequencing of the assembled products across different variants showed over 80% of the colonies were correctly assembled and free of any mutations (Fig. 1C). Nanopore sequencing of the entire BAC construct was achieved in ~20 h with >250x coverage (Supplementary Fig. 2b) and results were verified by NGS (Supplementary Fig. 3A). No mutations were present in any plasmids used in this study; this is consistent with the reported stability and reliability of BAC vectors[29]. Of note, the assembled plasmid can be induced to high copy number replication (>1 mg/L of bacterial culture) by addition of arabinose and in a confirmatory digest showed all expected digestion products (Fig. 1D). We confirmed that the assembled plasmid serves as template for in vitro transcription of full-length viral RNA, seen by co-migration of the RNA band with the template DNA (Fig. 1E). Of note, the HiScribe kit was faster in

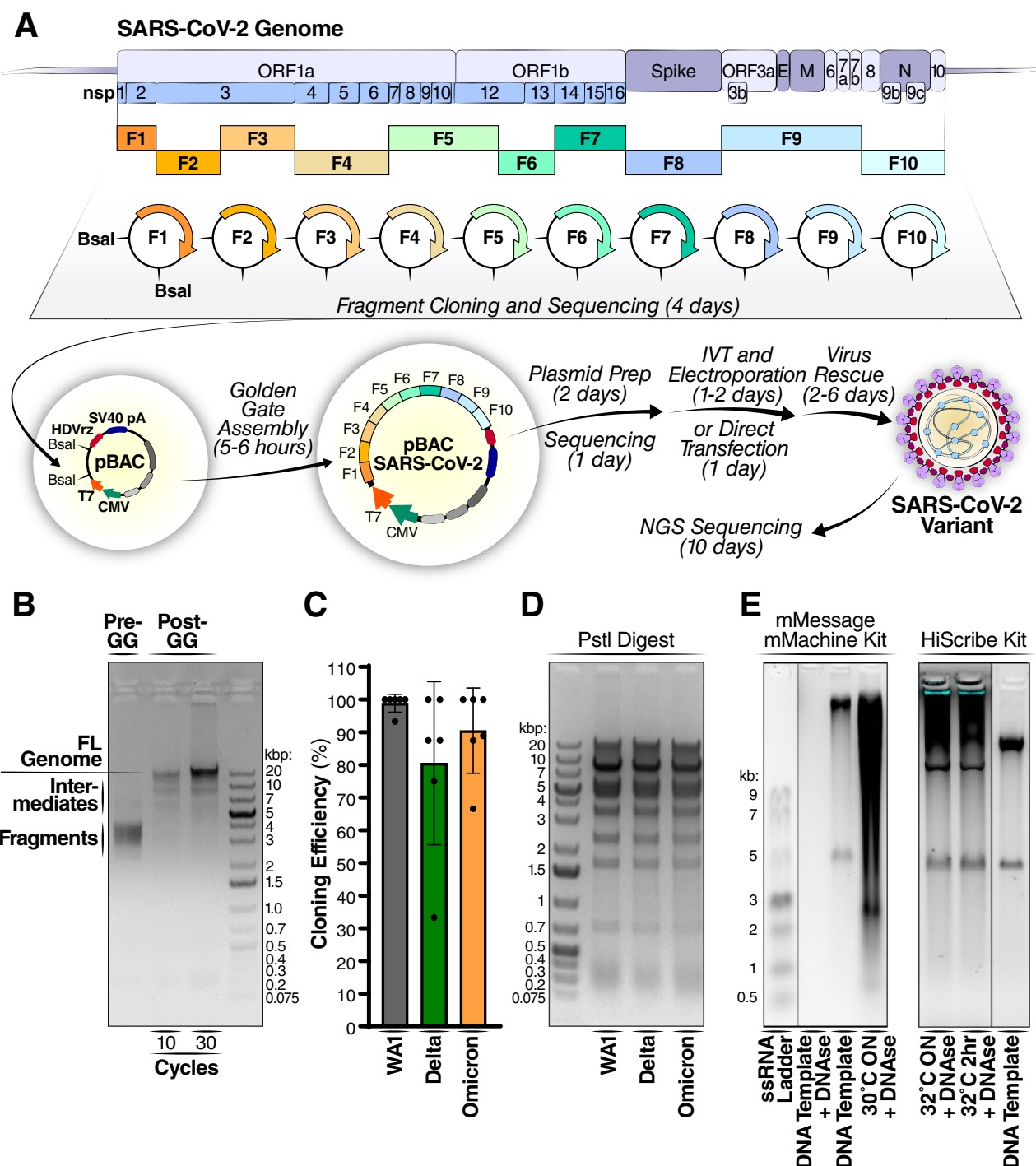

**Fig. 1 | Golden gate assembly enables rapid cloning of SARS-CoV-2 variants.**
**A** Schematic of cloning methodology and generation of infectious clones. The viral genome was rationally divided into 10 fragments and assembled into a bacterial artificial chromosome (BAC) vector containing T7 and cytomegalovirus (CMV) promoters, hepatitis delta virus ribozyme (HDVrz), and simian virus 40 (SV40) polyA sequence. The assembled vector was then directly transfected into cells or first in vitro transcribed into RNA, followed by electroporation into cells to generate SARS-CoV-2 variants. The estimated time required for each step is indicated in parentheses. IVT, in vitro transcription. The Electroporation image was created with Biorender. **B** Agarose gel electrophoresis of Golden Gate (GG) assembly of the 10 fragments. The gel is representative of 3 independent assembly reactions. **C** Cloning efficiency of

SARS-CoV-2 variant infectious clones. Correct colonies are defined as those with perfectly correct sequence across the entire genome. Data are shown as average ± SD of 6 independent cloning experiments. A total of 47 (range 3–15), 37 (range 3–12), and 43 (range 3–15) colonies were analyzed for the WA1, Delta, and Omicron variants, respectively. **D** Agarose gel electrophoresis of PstI digest of 0.5 μg of SARS-CoV-2 variant infectious clone plasmids, demonstrating high quantity and quality of plasmid preps. The gel is representative of 5 independent plasmid preps and PstI digests. **E** In vitro transcription of assembled plasmid to generate full-length RNA under different conditions with two different commercial kits. The gels are representative of at least 2 independent IVT reactions for each kit. ssRNA, single-stranded RNA; ON, overnight. Source data are provided as a Source Data file.

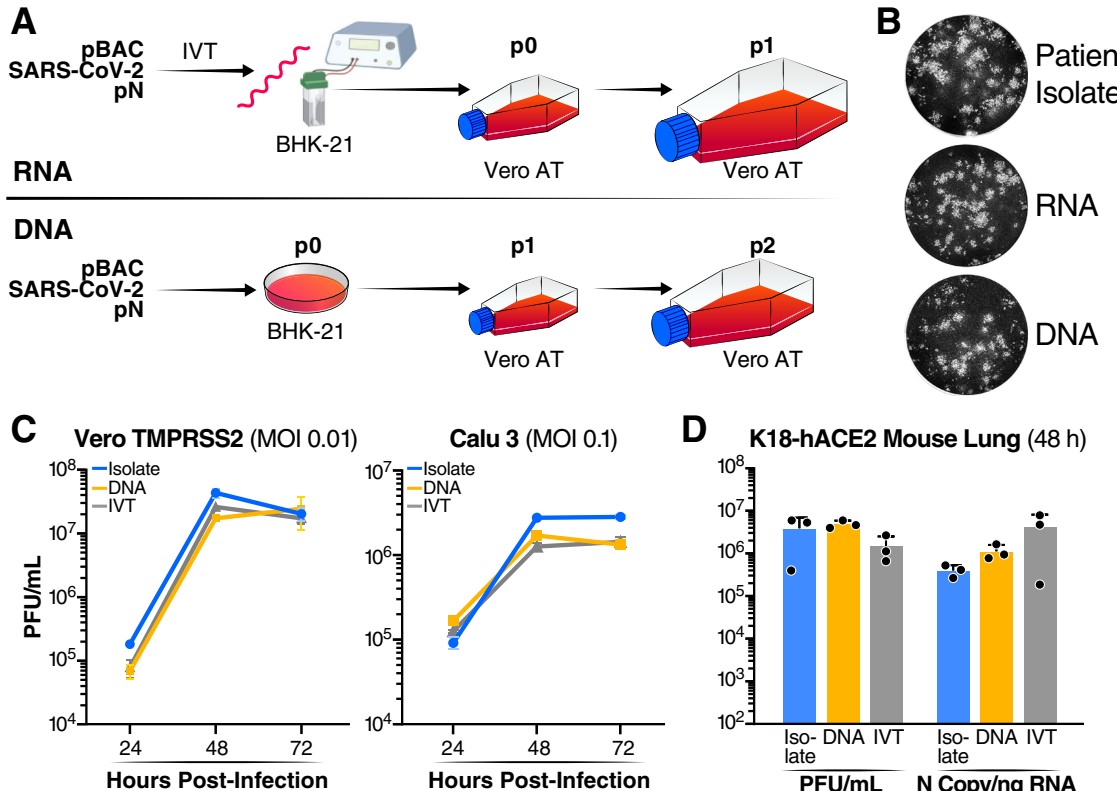

**Fig. 2 | DNA- and RNA-launched viruses replicate similarly to virus derived from patient isolate. A** Schematic of virus rescue from RNA or DNA. For RNA-launched virus rescue, in vitro transcribed RNA from viral construct (pBAC SARS-CoV-2) and N expression construct (pN) is electroporated into BHK-21 cells followed by co-culture with Vero ACE2 TMPRSS2 (Vero AT) cells to yield p0 viral stock, which is propagated in the same cells onward. For DNA-launched virus rescue, pBAC SARS-CoV-2 and pN are directly transfected into BHK-21 cells to yield p0 viral stock, which is then propagated in Vero AT cells. **B** Plaque morphology of DNA- and RNA-launched and patient-derived Delta variant viruses. Images were pseudocolored to black and white for optimal visualization. The images represent three independent replicate experiments. **C** Growth kinetics of the viruses in B in Vero TMPRSS2 and Calu3 cells over 72 h as measured by infectious particle release by plaque assay. Average of three independent experiments analyzed in duplicate ± SD is shown. **D** Replication of the viruses in B was assessed in K18-hACE2 mice lungs at 48 h post-infection by infectious particle release by plaque assay and viral RNA by RT-qPCR. Data are presented as average ± SD of 3 mice in each group. Source data are provided as a Source Data file.

producing the full-length RNA than the mMessage mMachine kit (2 h vs overnight reaction, respectively), but yielded less total RNA (10 μg/reaction vs >100 μg/reaction, respectively).

Cloning of a full-length variant from sequence to sequenced plasmid using pGLUE can be achieved on average in 1 week (Fig. 1A for average timeline). The assembled DNA construct can then be transfected directly into appropriate target cells for recovery of infectious virus or can be first transcribed into RNA with T7 polymerase followed by electroporation into cells and virus rescue (Fig. 2A). We did not observe any consistent differences in viruses launched from DNA or RNA and usually transfect the plasmid DNA directly (Fig. 2B–D). We further compared a cloned Delta variant, either RNA- or DNA-launched, with a Delta patient isolate in cell culture and animal models of infection. The patient-derived and de novo constructed recombinant viruses were sequence verified (Supplementary Fig. 3B-D), had the same plaque morphology (Fig. 2B), similar replication kinetics in Vero TMPRSS2 and Calu3 cells (Fig. 2C), and produced similar viral loads in K18-hACE2 mice after intranasal inoculation (Fig. 2D). Thus, the pGLUE method is robust and produces viruses that are comparable to patient-derived viruses.

### Omicron mutations in Spike and ORF1ab reduce viral particle production and intracellular RNA levels

Using pGLUE, two recombinant clones of the Delta and Omicron variants were constructed (Fig. 3A). For the Delta and Omicron variants, the mutations selected were representative of >90% of all Delta and Omicron sequences on the GISAID database as of January 2022. In addition, we focused on two naturally occurring viruses: (1) "Deltacron" which harbors the Omicron Spike ORF within the Delta variant[40–42] and (2) a virus harboring the Omicron ORF1ab within the Delta variant also found in the GISAID database (Supplementary Fig. 4). Full-length genomes were constructed using pGLUE and labeled Delta-Omicron S and Omicron-Delta, respectively (Fig. 3A). The resulting viruses were propagated in Vero ACE2 TMPRSS2 cells, and infectious particle production was measured by plaque assay (Fig. 3B).

Significant differences in plaque morphology were observed (Fig. 3B). The Delta variant produced the largest plaque sizes of the tested viruses while plaques produced by Omicron were the smallest. Similar data were recently reported for Delta and Omicron Spike and point to the Omicron Spike RBD as the mediator of the smaller plaque size[43]. Delta-Omicron S produced small plaques, which were slightly larger than that of the Omicron variant. This indicates that receptor binding and fusion capabilities are largely endowed by the Spike protein and that the Omicron Spike protein has reduced fusogenic properties compared to Delta's. Interestingly, Omicron-Delta produced smaller plaques than the Delta variant pointing to negative contributions of the Omicron ORF1ab to this phenotype.

Next, the growth kinetics of the different viruses were determined at 24, 48 and 72 h in Calu3 cells infected at a multiplicity of infection (MOI) of 0.1 (Fig. 3C, D). Of note, the presence of the Omicron Spike ORF in the Delta variant attenuated particle production significantly. This confirms that Spike mutations play a significant role in tuning

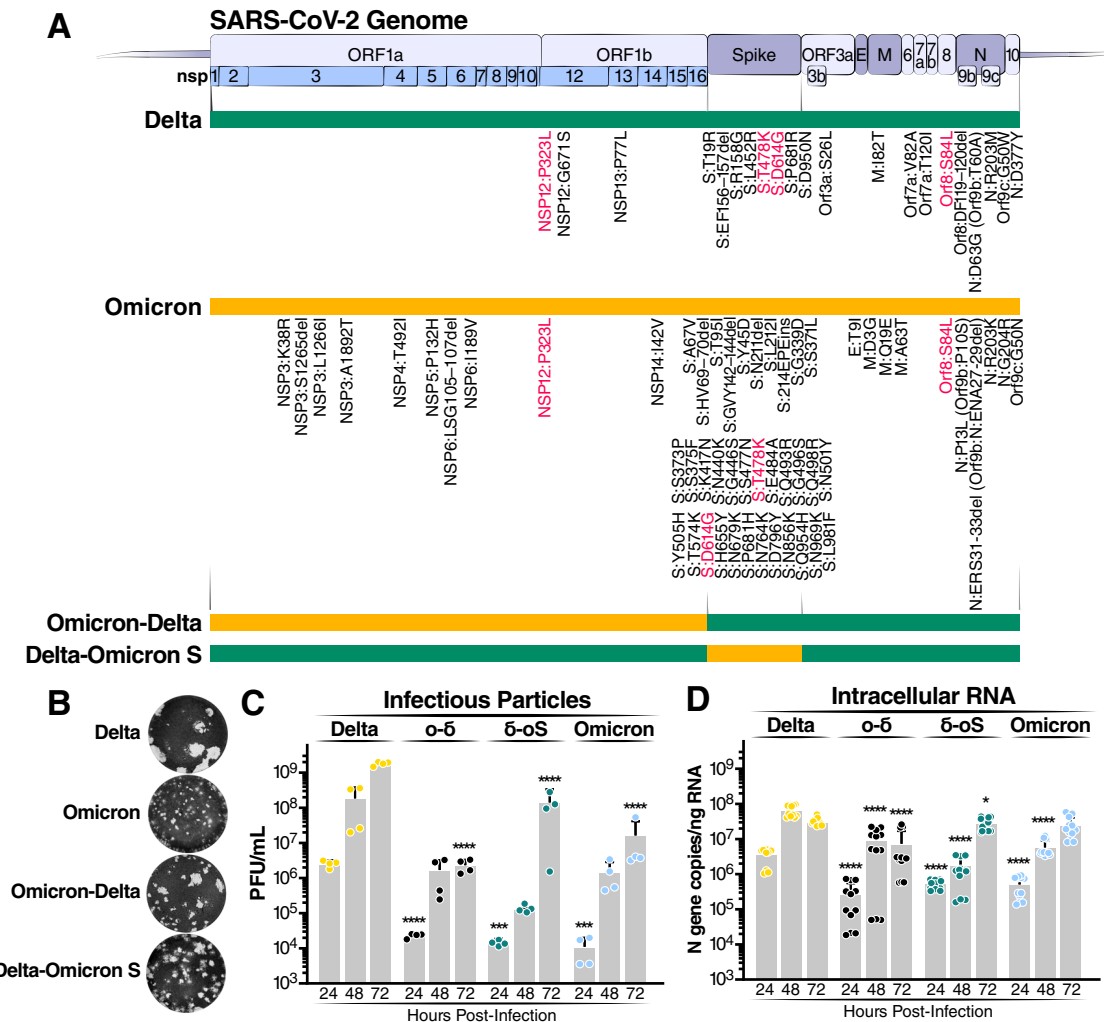

**Fig. 3 | Omicron mutations in Spike and ORF1ab reduce viral particle production and intracellular RNA levels. A** Schematic of recombinant infectious clones of Delta (green) and Omicron (orange) variants with indicated mutations. Mutations represent >90% of GISAID sequences of each variant as of January 2022. **B** Representative images of plaques from indicated recombinant infectious clones. Images were pseudocolored to black and white for optimal visualization. The images represent three independent replicate experiments. **C** Extracellular infectious particles from infected Calu3 cells (MOI 0.1). Average of four independent experiments analyzed in duplicate ± SD is shown and compared to Delta by two-sided Student's *T*-test at each timepoint. **D** Intracellular RNA was quantified from infected Calu3 cells (MOI of 0.1). Data are expressed in absolute copies/ng based on a standard curve of N gene with known copy number. Average of four independent experiments analyzed in triplicate (independent qPCR runs) ± SD is shown and compared to Delta by two-sided Student's *T*-test at each timepoint. O-δ, Omicron-Delta recombinant; δ-oS, Delta-Omicron S recombinant. *$p < 0.05$; ***$p < 0.001$; ****$p < 0.0001$ by two-sided Student's *T*-test. Source data are provided as a Source Data file.

Omicron's replicative fitness[43–45]. However, the presence of Omicron ORF1ab in Delta also significantly reduced infectious particle production, indicating that mutations in ORF1ab contribute to Omicron attenuation. The same was observed when intracellular RNA levels were determined by RT-qPCR (Fig. 3D). Collectively, these data indicate that mutations in Spike and ORF1ab contribute to reduced viral fitness of the Omicron variant in cell culture.

## Spike-independent attenuation of Omicron

To define further Spike-independent differences between Omicron and Delta, a replicon system lacking the Spike protein was constructed (Fig. 4A, B). This system does not produce viral particles unless Spike is provided in trans, allowing only a single round of infection. Briefly, the entire Spike coding sequence was replaced with the one for secreted nanoluciferase (nLuc) and enhanced green fluorescent protein (EGFP). Of note, we used only the luciferase readout in this study because of its high sensitivity and dynamic range. Transfection of the replicon construct successfully launches viral genome replication in transfected

cells as indicated by detectable luciferase activity in the cell supernatant (Fig. 4C). Interestingly, the Delta replicon produced fivefold higher luciferase signal than the Omicron replicon (Fig. 4C), underscoring that non-Spike mutations are contributing to Omicron attenuation. No significant luciferase activity was observed when the supernatant from these cultures was transferred to permissive cells (Fig. 4D), confirming the absence of infectious particle production from the transfected replicon construct. When the appropriate Spike vector was cotransfected with the replicon construct, production of infectious particles occurred as indicated by luciferase activity in both transfected and infected cells (Fig. 4C, D).

Surprisingly, transfection of increasing amounts of the Spike expression construct while maintaining a constant amount of the replicon construct led to increasing luciferase activity in both transfected and infected cells (Fig. 4C, D). Previous reports on particle assembly using only viral structural proteins suggested that only trace amounts of Spike are necessary for particle assembly and that higher amounts led to lower particle assembly[46,47]. This indicates that other

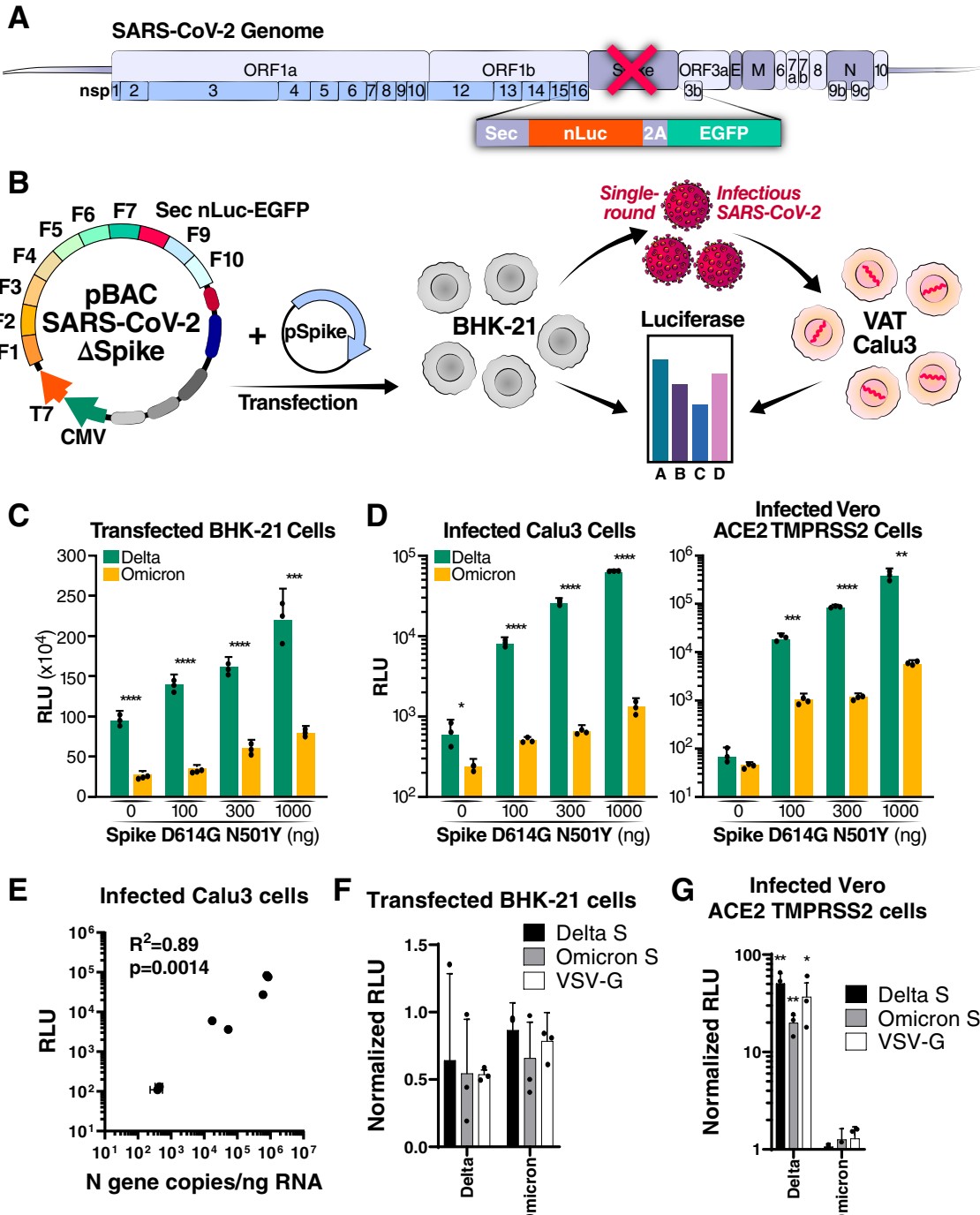

**Fig. 4 | Omicron mutations attenuate viral replication independent of Spike.**
**A** Schematic of the replicon system in which the Spike gene was replaced with secreted Nanoluciferase (Sec nLuc) and enhanced green fluorescent protein (EGFP) separated by a self-cleaving P2A peptide. **B** Experimental workflow of the SARS-CoV-2 replicon assay. VAT, Vero cells stably overexpressing ACE2 and TMPRSS2. **C** Luciferase readout from cells transfected with increasing amounts of Spike expression construct paired with either the Delta or Omicron replicon plasmids. Average of three independent experiments analyzed in duplicate ± SD and pairwise comparisons between the Delta and Omicron variants by two-sided Student's $T$-test are shown. **D** Luciferase readout from Calu3 or Vero ACE2 TMPRSS2 cells infected with supernatant from BHK-21 cells transfected with Delta or Omicron replicons in B. Shown are the average of three independent experiments analyzed in duplicate ± SD and pairwise comparisons between the Delta and Omicron variants by two-sided

Student's $T$-test. **E** Correlation analysis of replicon-generated relative luminescence unit (RLU) signal in the supernatant of infected Calu3 cells with abundance of viral N gene RNA in the same well as measured by RT-qPCR. Pearson's correlation (two-tailed) was utilized to calculate $R^2$ value and $p$ value. **F** Luciferase readout from transfected BHK-21 with Delta and Omicron replicons and a Delta Spike, Omicron Spike, or VSV-G expression vectors. The Omicron replicon plasmid was transfected at twice the amount of the Delta replicon. Average of three independent experiments analyzed in duplicate ± SD is shown. **G** Luciferase readout from infected Vero ACE2 TMPRSS2 cells with supernatant from F. Average of three independent experiments analyzed in duplicate ± SD is shown, and pairwise comparisons were made relative to the Omicron variant by two-sided Student's $T$-test. *$p < 0.05$; **$p < 0.01$; ***$p < 0.001$; ****$p < 0.0001$ by two-sided Student's $T$-test. Source data are provided as a Source Data file.

viral proteins, which were not present in these previous experiments, are important in Spike processing or mediate critical steps in the assembly process. Regardless of the Spike amount transfected, the Omicron variant consistently performed worse, as shown by reduced luciferase signal, compared with the Delta variant, in both transfected and infected cells (Fig. 4C, D). This demonstrates attenuation of the Omicron variant is at the RNA replication step.

We performed several confirmatory experiments to validate the luciferase readout of the replicon system: (i) we infected Calu3 cells with serial dilutions of replicon-generated viral particles and measured luciferase activity at 72 h after infection in the supernatant as well as N gene copies in infected cells by quantitative RT-PCR. Luciferase activity correlated highly with N gene copies, underscoring the validity of the reporter assay (Fig. 4E, Pearson's $R^2 = 0.89$, $p = 0.0014$). (ii) To ascertain that viral particles were produced by transfected cells and caused luciferase production after infection, we pelleted particles from supernatant of transfected cells by ultracentrifugation over a sucrose cushion (Supplementary Fig. 5A). Subsequent infection of Vero ACE2 TMPRSS2 cells demonstrated that the infectious agents were in the pellet, and not the supernatant, of the ultracentrifuged material (Supplementary Fig. 5B). (iii) We doubled the amount of Omicron replicon plasmid to obtain equal luciferase values in transfected cells. In addition, we varied the type of envelope that was cotransfected with the replicon plasmid and included Delta Spike, Omicron Spike, or the universal vesicular-stomatitis virus (VSV) glycoprotein to assess their impact on replicon infectivity and RNA replication (Fig. 4F, G). After infection, the Omicron replicon consistently produced low luciferase signal across all viral envelopes despite adjusted RNA levels (Fig. 4G). These results confirm that viral RNA replication is attenuated in the Omicron variant independently from Spike.

## Omicron NSP6 slows viral RNA replication

To map the contribution of non-Spike Omicron mutations on viral RNA replication within the Omicron genome, we constructed a series of Omicron replicons, in which viral proteins—individually or combined—were substituted with the corresponding proteins from Delta (Omicron-Delta). We only focused on proteins that contained mutations distinguishing Omicron from Delta. These replicon constructs were transfected along with Delta Spike and Nucleocapsid expression vectors to harvest virions for subsequent single-round infection experiments. Delta and Omicron replicons without substitutions were used as controls, and luciferase values in both transfected and infected cells were measured (Fig. 5A, B). Replacement of Omicron NSP6 with Delta's restored the luciferase signal in transfected BHK-21 cells and infected Vero ACE2 TMPRSS2 and Calu3 cells (Fig. 5A, B), indicating that NSP6 mutations contribute to Spike-independent attenuation of Omicron. Interestingly, replacement of NSP5 with that of Delta markedly reduced luciferase signal in all conditions (Fig. 5B) suggesting that the Omicron NSP5 mutation is associated with enhanced RNA replication compared to Delta. In vitro analysis of protease activity of Omicron's NSP5 has previously shown similar activity to that of Delta but reduced thermal stability[48]. Delta NSP13 slightly reduced luciferase activity while NSP14 enhanced activity in Vero ACE2 TMPRSS2 cells, but this effect was not observed in Calu3 cells (Fig. 5B). Similarly, substitution of structural E and M proteins increased luciferase in Calu3, but not Vero ACE2 TMPRSS2 cells, while Delta ORF8 and N substitution decreased it only in infected Calu3 cells. Conducting the experiment with Omicron, instead of Delta, Spike and Nucleocapsid expression constructs led to similar results (Supplementary Fig. 6A). These results point to multiple, possibly epistatic, interactions between nonstructural proteins causing Omicron attenuation, with the most consistent effect observed across all cell types mapping to NSP6 and double-membrane vesicle (DMV) formation, while the NSP5 protease evolved to support higher RNA replication in the context of the Omicron variant.

To independently validate the opposing trajectories of Omicron mutations in NSP5 and 6 on viral RNA replication, we performed the complimentary experiments by substituting Omicron NSP5 and 6 proteins, either individually or combined, in Delta replicons, including NSP4 as a control (Delta-Omicron). Omicron NSP6 within a Delta replicon consistently decreased RNA luciferase levels in transfected and infected cells, confirming that NSP6 of Omicron contributes to attenuation (Fig. 5C–D). Substituting Delta NSP5 with the one from Omicron increased luciferase levels in transfected, but not infected cells. Interestingly, combined insertion of Omicron NSP5 and 6 proteins into a Delta replicon, decreased luciferase levels, although slightly less than NSP6 alone, indicating that NSP6 function in Omicron dominantly contributes to attenuation (Fig. 5C–D). The same was observed when Delta NSP5 and 6 proteins combined were inserted into the Omicron replicon; the combined substitution increased RNA replication to similar levels as the NSP6 substitution alone, underscoring the dominant effect of NSP6 over NSP5 (Fig. 5C–D). The NSP4 recombinants did not show any difference compared with parental replicons as expected. Similar results were observed regardless if replicons were cotransfected with Delta (Fig. 5D) or Omicron (Supplementary Fig. 6B) Spike and Nucleocapsid expression vectors. These results demonstrate that Omicron mutations in NSP6 play a dominant role in attenuating viral RNA replication.

NSP6 connects DMVs to the ER through zippered ER connectors and decreases lipid droplet (LD) content in infected cells by allowing flow of lipids from the ER to DMVs[49]. To compare NSP6 function between Omicron and Delta, we transiently expressed each NSP6 protein as a FLAG-tagged version in HEK293T cells and stained cells using FLAG antibodies and LipidTox Deep Red for LDs. Delta NSP6-expressing cells showed significantly decreased intensity of LD staining compared to those transfected with Omicron NSP6, which were similar to cells transfected with the empty vector (Fig. 5E, F). This indicates that Delta, but not Omicron, NSP6 increases LD consumption, consistent with the decrease in LD staining intensity recently reported[49]. The same experiments were performed in cells infected with the chimeric replicons and stained for LDs and RNA replication using antibodies against double-stranded RNA[49]. In infected cells expressing the Delta NSP6 protein, the intensity of LD staining was consistently lower than in cells expressing Omicron NSP6, regardless of the variant genetic background, confirming that NSP6 function in Omicron is diminished (Fig. 5G, H). Collectively, these data support the model that Omicron mutations in NSP6 impair lipid flow to replication organelles and consequently reduce viral RNA replication.

## Discussion

Collectively, use of pGLUE and the ability to rapidly generate replicating viral genomes revealed that Omicron attenuation, in addition to Spike adaptation, is driven by decreases in RNA replication with lipid-regulatory functions of NSP6 playing a central role in the attenuation process. Our data provide both technical and biological advances. Technically, pGLUE is an optimized single-pot ligation system that allows molecular interrogation of entire SARS-CoV-2 genomes within weeks. Biologically, we dissected the contribution of each Omicron mutation across ORF1ab and found that previously unappreciated Omicron mutations in NSP6 lower viral fitness with a specific effect on LD consumption.

Generating molecular viral clones is important, given the delay with obtaining regionally occurring patient isolates, the risk of undesired mutations during prolonged viral propagation, and the existence of toxic sequences that limit standard molecular cloning strategies. Using pGLUE, we routinely design and produce the pBAC plasmid containing individual viral variant genomes within a week. This efficiency enables us to address real-world changes in viral evolution with respect to all lifecycle steps. pGLUE is different from previous methods[23,25,27–32] in that: (i) it employs rational fragment design where

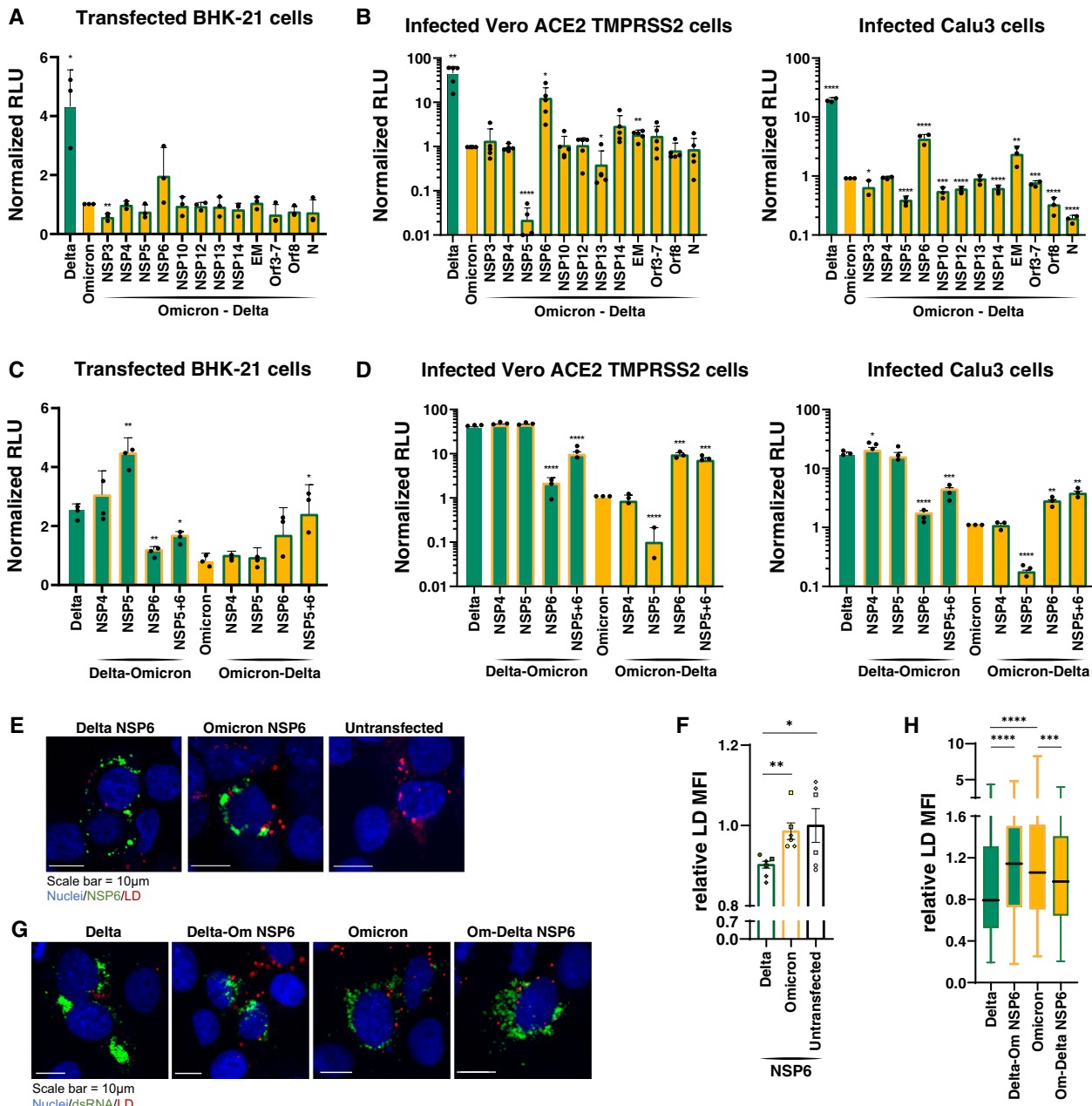

**Fig. 5 | Omicron NSP6 slows viral RNA replication. A** Luciferase readout from transfected BHK-21 with Delta, Omicron, and Omicron-Delta recombinants replicons as indicated and a Delta Spike and Nucleocapsid expression vectors. Average of three independent experiments analyzed in duplicate ± SD is shown, and pairwise comparisons were made relative to the Omicron variant by two-sided Student's *T*-test. **B** Luciferase readout from infected Vero ACE2 TMPRSS2 and Calu3 cells with supernatant from (**A**). Average of five and three independent experiments analyzed in triplicate and duplicate ± SD are shown for Vero ACE2 TMPRSS2 and Calu3 cells, respectively, and pairwise comparisons were made relative to the Omicron variant by two-sided Student's *T*-test. **C** Luciferase readout from transfected BHK-21 with Delta, Delta-Omicron recombinants, Omicron, and Omicron-Delta recombinants replicons as indicated and a Delta Spike and Nucleocapsid expression vectors. Average of three independent experiments analyzed in duplicate ± SD is shown, and pairwise comparisons were made relative to the Omicron variant by two-sided Student's *T*-test. **D** Luciferase readout from infected Vero ACE2 TMPRSS2 and Calu3 cells with supernatant from (**C**). Average of three independent experiments analyzed in duplicate ± SD is shown, and pairwise

comparisons were made relative to the Omicron variant by two-sided Student's *T*-test. **E** Representative images of transfected HEK293T cells with indicated FLAG-NSP6 expression vectors or untransfected control and stained for LD and FLAG. LD, lipid droplet. **F** Quantification of the relative LD mean fluorescent intensity (MFI) per transfected (BFP-positive) cells in images shown in (**E**). Average of six technical replicates ± SEM is shown, and pairwise comparisons were made as indicated by two-sided Student's *T*-test. **G** Representative images of infected Vero ACE2 TMPRSS2 cells with indicated replicons and stained for LD and dsRNA. dsRNA, double-stranded RNA. **H** Quantification of the relative LD mean fluorescent intensity (MFI) per dsRNA positive cells in images shown in G using box and whiskers plot, and pairwise comparisons were made as indicated by two-sided Student's *T*-test. Interquartile range (IQR) of boxplot is between 25th and 75th percentiles and center line indicates median value. Whiskers of boxplot is extended to the maxima and minima. Maxima is the largest value and minima is the smallest value in the dataset. \**p* < 0.05; \*\**p* < 0.01; \*\*\**p* < 0.001; \*\*\*\**p* < 0.0001 by two-sided Student's *T*-test. Source data are provided as a Source Data file.

each fragment contains distinct ORFs for rapid generation of recombinant chimeric viruses and replicons; (ii) it overcomes issues with toxic sequences in bacteria; (iii) the ligated fragments are cloned into a plasmid with high stability and reliability; and (iv) it takes full advantage of Golden Gate assembly to perform rapid single-pot ligation of the entire genome in less than six hours. We show here that the developed method is robust and can provide valuable insight into the molecular mechanisms of the SARS-CoV-2 lifecycle.

A large body of evidence has characterized the Omicron Spike protein and showed that it favors TMPRSS2-independent endosomal entry[9,50,51], has poor fusogenicity[51], and escapes neutralization by many antibodies[51–54]. Furthermore, studies using chimeric viruses bearing different Spike proteins showed that Spike is a major determinant of the Omicron attenuated replicative phenotype[43–45]. Our results with full-length molecular clones confirm these findings and underscore the critical role that the Spike protein plays in determining viral fitness and skewing viral adaptation towards immune escape.

Less work has been done so far to investigate the impact of the Omicron mutations outside of the Spike protein. Previously, a Spike-independent attenuation of the Omicron variant in animals has been reported[55]. Our data define a role of ORF1ab Omicron mutations, implicating reduced RNA replication and LD consumption with a potentially enhanced polyprotein processing capacity in the adaptation process. While our manuscript was under review, a study comparing ancestral (WA1) and Omicron chimeric molecular clones independently found an attenuating effect of Omicron NSP6[56]. NSP6 plays a critical role in mediating contact between DMVs and the ER membrane as well as channeling of lipids to viral replication organelles. Our data suggest that the Omicron mutations in NSP6 impair the LD channeling function of the protein. Further studies are needed to define the precise molecular consequences of these mutations, but it has recently been speculated that the Omicron mutation (LSG105-107del) lies within the largest ER luminal loop of the protein and a conserved *O*-glycosylation motif that can act as a spacer and may affect ER zippering activity[49].

NSP5 is a cysteine protease responsible for processing the viral polyprotein at sites between NSP4–16. There is one mutation in Omicron NSP5 (P132H), and our data indicate that it enhances viral RNA replication, but cannot compensate for decreased NSP6 function. The mutation lies between the catalytic domain and the dimerization domain of NSP5 and was shown to preserve protease activity or susceptibility to nirmatrelvir in vitro[48]. However, the mutation lowers the thermal stability of NSP5 in vitro. A possible explanation of our data is that the mutation affects the dimerization or the protease activity in the context of the polyprotein. Indeed, we observe an epistatic interaction between NSP5 and 6 where Delta NSP5 supports high levels of RNA replication in the presence of Delta NSP6 only, but not with Omicron NSP6.

Several studies have suggested that Omicron could have emerged due to epistatic interactions that may allow for the emergence of mutations not seen in other variants or that are very rare[57–59]. The low intra-host evolution for SARS-CoV-2 and relatively limited transmission bottleneck[60,61] suggest that Omicron may have evolved in chronically infected patients where the virus can cross through fitness valleys that may not be possible in an acute infection[57]. Interestingly, Omicron mutations in Spike (K417N and L981F) occur within conserved MHC-I-restricted CD8+ T-cell epitopes that may destabilize MHC-I complexes[62], indicating that T-cell immunity is an additional driver of SARS-CoV-2 evolution as in other viruses[63–65].

A potential advantage of our findings is that they may help generate candidates for live attenuated SARS-CoV-2 vaccines in the future[66]. A potential caveat is the introduction of antivirals such as nirmatrelvir, which targets specifically NSP5 and may drive development of selective resistance mutations[67–69]. SARS-CoV-2 continues to evolve, which carries the risk of reversion of the attenuating mutations

in Omicron. This is supported by recent reports on the enhanced infectivity and neutralization escape of Omicron-evolved subvariants[70–74]. Indeed, some recombinant viruses such as the BA.1 and BA.2 recombinant XE have a recombination point around the NSP5-6 junction disconnecting the two proteins and suggesting NSP6 as a potential evolutionary driver[75]. The ability to rapidly characterize full-length viral sequences will be increasingly valuable and will bring insight into the evolutionary path, viral fitness, expected pathogenicity as well as vaccine and antiviral medication responsiveness of emerging subvariants.

## Methods

### Ethics
All research conducted in this study complies with all relevant ethical regulations. All work conducted with replication-competent viruses was conducted in an approved biosafety level 3 (BSL3) laboratory and experiments approved by the Institutional Biosafety Committee of the University of California, San Francisco and Gladstone Institutes. All protocols concerning animal use were approved (AN169239-01C) by the Institutional Animal Care and Use committees at the University of California, San Francisco and Gladstone Institutes and conducted in strict accordance with the National Institutes of Health Guide for the Care and Use of Laboratory Animals.

### Cells
BHK-21 and HEK293T cells were obtained from ATCC and cultured in DMEM (Corning) supplemented with 10% fetal bovine serum (FBS) (GeminiBio), 1x glutamine (Corning), and 1x penicillin-streptomycin (Corning) at 37 °C, 5% $CO_2$. Calu3 cells were obtained from ATCC and cultured in AdvancedMEM (Gibco) supplemented with 2.5% FBS, 1x GlutaMax, and 1x penicillin-streptomycin at 37 °C and 5% $CO_2$. Vero cells stably overexpressing human TMPRSS2 (Vero TMPRSS2) (gifted from the Whelan lab[76]), were grown in DMEM with 10% FBS, 1x glutamine, 1x penicillin-streptomycin at 37 °C and 5% $CO_2$. Vero cells stably co-expressing human ACE2 and TMPRSS2 (Vero ACE2 TMPRSS2) (gifted from A. Creanga and B. Graham at NIH) were maintained in Dulbecco's Modified Eagle medium (DMEM; Gibco) supplemented with 10% FBS, 100 µg/mL penicillin and streptomycin, and 10 µg/mL of puromycin at 37 °C and 5% $CO_2$.

### Infectious clone preparation
To enable this rapid cloning strategy, the SARS-CoV-2 genome was divided into 10 fragments that correspond to different coding regions of the genome (Supplementary Fig. 2). The fragments were cloned into a pUC19-based vector with the bidirectional tonB terminator upstream and the T7Te and rrnB T1 terminators downstream of the SARS-CoV-2 sequence. All plasmids were sequenced using the Primordium Labs whole plasmid sequencing service. Prior to assembly, the fragments were PCR amplified and cleaned. To enable assembly of the full-length SARS-CoV-2 genome using BsaI-mediated Golden Gate assembly, the two BsaI sites in the genome (WA1 nt 17966 and nt 24096) were eliminated by introducing the following synonymous mutations (WA1 nt C17976T and nt C24106T) in fragments F6 and F8, respectively. The pBAC vector that can handle the full-length genome was purchased from Lucigen (cat # 42032-1). This vector was modified to include a CMV promoter, T7 promoter, BsaI sites, an HDVrz and SV40 polyA. The BsaI site at nt 2302 was mutated (C2307T) to allow use in the BsaI-mediated Golden Gate assembly. For the Golden Gate assembly, the 10 fragments and the pBAC vector were mixed in stoichiometric ratios in 1x T4 DNA ligase buffer (25 µL reaction volume). To the mixture was added BsaI-HF v2 (1.5 µL) and Hi-T4 DNA ligase (2.5 µL). The assembly was performed as follows in a thermal cycler: 30 cycles of 37 °C for 5 min, followed by 16 °C for 5 min. Then the reaction was incubated at

37 °C for 5 min and 60 °C for 5 min. 1 μL of the reaction was electroporated into EPI300 cells and plated onto LB + chloramphenicol plates and grown at 37 °C for 24 h. Colonies were picked and cultured in LB30 medium + 12.5 μg/mL of chloramphenicol for 12 h at 37 °C. 1 mL of the culture was diluted to 100 mL of LB30 medium + 12.5 μg/mL of chloramphenicol for 3–4 hours. The culture was diluted again to 400 mL of LB30 medium + 12.5 μg/mL of chloramphenicol + 1x Arabinose induction solution (Lucigen) for overnight. The pBAC infectious clone plasmid was extracted and purified using NucleoBond Xtra Maxi prep kit (Macherey-Nagel). The plasmid was then sequenced using Primordium Labs "Large" whole plasmid sequencing service. All plasmids constructed in the study will be available via Addgene.

### In vitro transcribed RNA preparation
20 μg of the pBAC infectious clone plasmid was digested with SalI and SbfI for at least 3 h at 37 °C in a 50-μL reaction. The digest was diluted to 500 μL with DNA lysis buffer (0.5% SDS, 10 mM Tris pH 8, 10 mM EDTA, and 10 mM NaCl) and 5 μL of proteinase K was added. The mixture was incubated at 50 °C for 1 h. The DNA was extracted with phenol and precipitated with ethanol. 2 μg of digested DNA was used to set up the IVT reactions according to the manufacturer's instructions for both the HiScribe and the mMessage mMachine kits except for the incubation times as indicated (Fig. 1E). The mMessage mMachine Kit was used to generate the RNA for all infectious clone experiments. After the IVT reaction, the RNA was extracted with RNA-STAT-60 and precipitated with isopropanol, according to the manufacturer's instructions. To generate N IVT RNA, the exact procedure above was followed, except that the plasmid was digested with SalI only and the IVT reaction was run for 2 h at 37 °C.

### Infectious clone virus rescue
To generate the RNA-launched SARS-CoV-2, the purified infectious clone RNA (10 μg) was mixed with N RNA (5 μg) and electroporated into $5 \times 10^6$ BHK-21 cells. The cells were then layered on top of Vero ACE2 TMPRSS2 cells in a T75 flask (Fig. 2A). After development of cytopathic effect, the virus was propagated onto Vero ACE2 TMPRSS2 to achieve high titer. To generate the DNA-launched SARS-CoV-2, the pBAC SARS-CoV-2 construct was directly cotransfected with N expression construct into BHK-21 cells in six-well plate (Fig. 2A). After 3 days post-transfection, the supernatant was collected and used to infect Vero ACE2 TMPRSS2 cells and passaged further to achieve high titer. All viruses generated and/or utilized in this study were NGS verified using the ARTIC Network's protocol[77].

### SARS-CoV-2 replicon assay
Plasmids harboring the full SARS-CoV-2 sequence except for Spike (1 μg) were transfected into BHK-21 cells along with Nucleocapsid and Spike expression vectors (0.5 μg each) in 24-well plate using X-tremeGENE 9 DNA transfection reagent (Sigma Aldrich) according to manufacturer's protocol. The supernatant was replaced with fresh growth medium 12–16 h post-transfection. The supernatant containing single-round infectious particles was collected and 0.45 μm-filtered 72 h post- transfection. The supernatant was subsequently used to infect Vero ACE2 TMPRSS2 cells (in 96-well plate) or Calu3 cells (in 24-well plate). The medium was refreshed 12–24 h post-infection. To measure luciferase activity, an equal volume of supernatant from transfected cells or infected cells was mixed with Nano-Glo luciferase assay buffer and substrate and analyzed on an Infinite M Plex plate reader (Tecan).

### SARS-CoV-2 virus culture and plaque assay
SARS-CoV-2 variants B.1.617.2 (BEI NR-55611) were propagated on Vero ACE2 TMPRSS2 cells, sequence verified, and were stored at −80 °C until use. For plaque assays, tissue homogenates and cell supernatants were analyzed for viral particle formation for in vivo and in vitro experiments, respectively. Briefly, Vero ACE2 TMPRSS2 cells were plated and rested for at least 24 h. Serial dilutions of inoculate of homogenate or supernatant were added on to the cells. After the 1-hour absorption period, 2.5% Avicel (Dupont, RC-591) was overlaid. After 72 h, the overlay was removed, the cells were fixed in 10% formalin for one hour, and stained with crystal violet for visualization of plaque formation.

### Analysis of viral sequences
Viral sequences were downloaded from the GISAID database and analyzed for mutations utilizing the Geneious Prime software version 2022.2.1. The GISAID mutation analysis tool was utilized to quickly filter for recombinants containing specific mutations prior to download.

### Real-time quantitative polymerase chain reaction (RT-qPCR)
RNA was extracted from cells, supernatants, or tissue homogenates using RNA-STAT-60 (AMSBIO, CS-110) and the Direct-Zol RNA Miniprep Kit (Zymo Research, R2052). RNA was then reverse-transcribed to cDNA with iScript cDNA Synthesis Kit (Bio-Rad, 1708890). qPCR reaction was performed with cDNA and SYBR Green Master Mix (Thermo Fisher Scientific) using the CFX384 Touch Real-Time PCR Detection System (Bio-Rad). N gene primer sequences are: Forward 5′ AAATTTTGGGGACCAGGAAC 3′; Reverse 5′ TGGCACCTGTGTAGGT CAAC 3′. The tenth fragment of the infectious clone plasmid was used as a standard for N gene quantification by RT-qPCR.

### K18-hACE2 mouse infection model
Mice were housed in a temperature- and humidity-controlled pathogen-free facility with 12-hour light/dark cycle and *ad libitum* access to water and standard laboratory rodent chow. Briefly, the study involved intranasal infection ($1 \times 10^4$ PFU) of 6–8-week-old female K18-hACE2 mice with Delta (DNA, RNA, and patient isolate). A total of 5 animals were infected for each variant and euthanized at 2 days post-infection. The lungs were processed for further analysis of virus replication.

### Cellular infection studies
Calu3 cells were seeded into 12-well plates. Cells were rested for at least 24 h prior to infection. At the time of infection, medium containing viral inoculum was added on the cells. One hour after addition of inoculum, the medium was replaced with fresh medium. The supernatant was harvested at 24, 48, and 72 h post-infection for downstream analysis.

### Staining for LDs in transfected and infected cells
HEK293T cells were transfected with NSP6 expression vector gifted from the Krogan lab[78], and modified with mTagBFP2 in 6-well plates. 48 h after transfection, cells were washed with PBS, lifted with trypsin, and plated onto poly-L-lysine treated 24-well glass-bottom plates (Corning). The cells were incubated overnight, the culture medium was removed, cells were fixed with 4% paraformaldehyde, permeabilized with 0.1% TritonX-100 in PBS, and probed for Mouse-anti-FLAG M2 (1:200) and Donkey-anti-Mouse-AlexaFluor 488 (1:200). Cells were stained with LipidTox Deep Red (1:500) and Hoechst (1:500) in Hank's Balanced Salt Solution (HBSS), washed with HBSS, and resuspended in HBSS for imaging. Transfected cells were imaged on an Olympus FV3000RS confocal microscope with a 40X objective, and LD fluorescence quantified using Imaris 9.9.1 software.

Vero ACE2 TMPRSS2 cells ($3 \times 10^4$) were infected with replicons in 96-well optical plastic and incubated overnight. The culture medium was removed and replaced with LipidTox Deep Red (1:500) and Hoechst (1:500) in HBSS, washed with HBSS, and imaged in Live Cell Imaging Buffer (Invitrogen). For quantification, cells were imaged and

analyzed on a ImageXpress Micro confocal microscope (Molecular Devices) with a 10X objective and a custom analysis program for GFP and LD intensity. $2 \times 10^5$ Vero ACE2 TMPRSS2 cells were infected with replicons in 24-well glass-bottom plates and incubated overnight. The culture medium was removed, cells were fixed with 4% paraformaldehyde, permeabilized with 0.1% TritonX-100 in PBS, and probed for Mouse-anti-dsRNA J2 (1:200) and Donkey-anti-Mouse-AlexaFluor 488 (1:200). Cells were stained with LipidTox Deep Red (1:500) and Hoechst (1:500) in HBSS, washed with HBSS, and resuspended in HBSS for imaging. For higher resolution images, the 24-well plate was imaged on an Olympus FV3000RS confocal microscope with a 40X objective.

The origin of the antibodies are as follows:

Anti-FLAG: Sigma Aldrich F1804, Mouse, M2 clone; Anti-dsRNA: Cell Signaling 76651, Mouse, J2 clone; Anti-Mouse AlexaFluor 488: ThermoFisher A-21202, Donkey, polyclonal.

## Statistics & reproducibility

No statistical method was used to predetermine sample size. All experiments were repeated at least three times and no data were excluded from the analyses. The experiments were not randomized and the investigators were not blinded to allocation during experiments and outcome assessment. All statistical analyses and figure rendering was conducted with GraphPad PRISM version 9.4.1 (GraphPad Software, San Diego, California USA, www.graphpad.com).

## Reporting summary

Further information on research design is available in the Nature Portfolio Reporting Summary linked to this article.

## Data availability

All data supporting the findings of the present study are available in the article, extended data and supplementary figures, or are available from the corresponding authors on request. Source data are provided with this paper.

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

## Acknowledgements

We thank the Whelan laboratory for providing the Vero cells over-expressing human TMPRSS2 and A. Creanga and B. Graham for the Vero cells overexpressing human ACE2 and TMPRSS2. We acknowledge funding support from the NIH F31 AI164671-01 (IPC). We gratefully acknowledge support from the Roddenberry Foundation, P. and E. Taft and the Pendleton Foundation (MO). M.O. is a Chan Zuckerberg Biohub – San Francisco Investigator.

## Author contributions

Conceptualization: T.Y.T. and M.O. Investigation: T.Y.T., I.P.C., J.M.H., T.T., K.W., G.R.K., A.M.S., A.C., R.K.S., H.S.M., B.H.B., C.L.T., M.M., M.M.K.,

B.K.S., and G.R.K. Methodology: T.Y.T., I.P.C., J.M.H., T.T., and M.O. Supervision: S.W., J.A.D., and M.O. Writing: T.Y.T., I.P.C., J.M.H., and M.O.

## Competing interests

The authors declare the following competing interests: T.Y.T., T.T., and M.O. are inventors on a patent application filed by the Gladstone Institutes that covers the use of pGLUE to generate SARS-CoV-2 infectious clones and replicons. All other authors declare no competing interests.
