## [Peer Review File · Nature Communications]

Reviewers' Comments:

Reviewer #1:

Remarks to the Author:

While I am still not convinced that pGLUE system provides “revolutionary breakthrough” in reverse genetics for SARS-CoV-2 I believe that it does have some attractive features which make it useful for researchers and worthy of publication.

Most of my major concerns were addressed and authors provided new data and experimental details which improved the manuscript.

Importantly, authors now provide new data (new Fig 5) which show functional significance of Omicron NSP6 in inhibiting lipid droplet redistribution from cytoplasm to membrane sites of viral RNA replication as the main Spike-independent determinant of Omicron attenuation. The function of NSP6 in redistribution of lipid droplets was shown previously and NSP6 has been recently shown in a Nature paper to contribute to Omicron attenuation. However, mechanism for this attenuative effect was not investigated. Data in this manuscript now provide mechanistic insight into the link between Omicron NSP6-mediated change in redistribution of lipid droplets away from RNA replication sites and attenuation of Omicron RNA replication.

Some relatively minor issues, however, still require attention:

The claims on improved (optimised) protease activity of Omicron NSP5 is not supported by the data in this manuscript (no protease activity is actually shown) and also by previously published data. Therefore, it should be either removed or significantly toned down in Results and Discussion.

Technically, the method still requires cloning of 10 individual fragments. While some of them that do not encode any differences or only 1-2 differences between variants can be used for assembly of new variants or chimeric viruses, cloning of other fragments containing substantial number of differences between variants will still be required. Four day's time frame claimed to be sufficient for this step is rather optimistic. Also, while authors claim that high copy number plasmids with all cloned fragments are stable, no data on their stability or in fact also the stability of full-length BAC clones are actually presented. It'd be good to see the stability data and corresponding DNA yields.

The statements on “rational design” of fragments and “containing distinct ORFs” are overused in the manuscript and are not strictly correct as most fragments except fragment 8 encode multiple ORFs and some fragments split ORFs (e.g. fragments 5 and 6 join in the middle of ORF12). CPER fragments can also be easily designed with any particular viral gene/gene combinations in mind. The authors may want to limit the number of times this phrase is mentioned and/or tone it down.

Statement that CPER was adapted from previous tick-borne virus research citing 1995 paper which used assembly of full-length viral cDNA genome under SP6 promoter from 2 PCR fragments is not strictly correct as most CPER publications employ CMV promoter-based assembly which does not require an in vitro RNA transcription step and also utilise multiple fragment assembly into a circular DNA. I also don't see how this statement and similar statement for in vitro ligation system are enhancing/proving novelty of the pGLUE system as this system is essentially an adaptation of Golden Gate assembly.

It is also not clear why authors state that CPER has limited capacity for introducing new mutations. It has been shown that CPER can easily utilise fragments cloned in plasmid vectors in addition to the fragments amplified directly from viral cDNA, which is similar to the pGLUE system. In fact, most recent CPER (CPEC) publication (doi: 10.1128/spectrum.03385-22) devised a neat approach for simultaneous introduction of multiple mutations in multiple fragments. Perhaps authors should

consider removing or rephrasing this statement.

We appreciate the reviewer finding our study “useful for researchers and worthy of publication”. Below are point-by-point responses to the remaining comments:

Reviewer #1 (Remarks to the Author):

The claims on improved (optimised) protease activity of Omicron NSP5 is not supported by the data in this manuscript (no protease activity is actually shown) and also by previously published data. Therefore, it should be either removed or significantly toned down in Results and Discussion.

We have removed the protease activity comments in the results section and now write that the “Omicron NSP5 mutation is associated with enhanced RNA replication” at Line 251, which is supported by data in the manuscript (Fig. 5B).

Technically, the method still requires cloning of 10 individual fragments. While some of them that do not encode any differences or only 1-2 differences between variants can be used for assembly of new variants or chimeric viruses, cloning of other fragments containing substantial number of differences between variants will still be required. Four day’s time frame claimed to be sufficient for this step is rather optimistic. Also, while authors claim that high copy number plasmids with all cloned fragments are stable, no data on their stability or in fact also the stability of full-length BAC clones are actually presented. It’d be good to see the stability data and corresponding DNA yields.

We routinely clone up to 9 mutations per fragment using NEB HiFi DNA Assembly without issues and within a 3–4 day timeframe. This is the maximum number of mutations of emerging variants compared to the parental strain. Note: the number of mutations in spike for Omicron subvariants is typically larger relative to WA1, but not when cloned onto the parental Omicron BA.1 or BA.2 Spike fragment.

With regards to the stability and yield of BAC plasmids containing all fragments, we routinely culture and induce bacteria carrying the BAC plasmid in the lab and obtain on average >1mg of high-quality DNA per 1 L of culture, which is indicated at line 144 in the manuscript. In addition, each plasmid is quality checked by restriction enzyme digest to demonstrate equal abundance of all fragments in the plasmid and that no recombination has occurred (see Fig. 1D). From the 25 full length SARS-CoV-2 pBAC plasmids we have generated in this study, 0 was unstable or has led to poor yields. We have also presented representative sequencing coverage data in Supplemental Figure 2 demonstrating the high quality of the generated plasmids.

The statements on “rational design” of fragments and “containing distinct ORFs” are overused in the manuscript and are not strictly correct as most fragments except fragment 8 encode multiple ORFs and some fragments split ORFs (e.g. fragments 5 and 6 join in the middle of ORF12). CPER fragments can also be easily designed with any particular viral gene/gene combinations in mind. The authors may want to limit the number of times this phrase is mentioned and/or tone it down.

We have removed some of the statements to limit the number of times this phrase is mentioned (page 8, line 310).

Statement that CPER was adapted from previous tick-borne virus research citing 1995 paper

which used assembly of full-length viral cDNA genome under SP6 promoter from 2 PCR fragments is not strictly correct as most CPER publications employ CMV promoter-based assembly which does not require an in vitro RNA transcription step and also utilise multiple fragment assembly into a circular DNA. I also don't see how this statement and similar statement for in vitro ligation system are enhancing/proving novelty of the pGLUE system as this system is essentially an adaptation of Golden Gate assembly.

We have now clarified our statement in the introduction section to indicate that we are referring to the assembly using PCR and not the overall method of virus rescue, such as promoter choice (page 3, line 87).

It is also not clear why authors state that CPER has limited capacity for introducing new mutations. It has been shown that CPER can easily utilise fragments cloned in plasmid vectors in addition to the fragments amplified directly from viral cDNA, which is similar to the pGLUE system. In fact, most recent CPER (CPEC) publication (doi: 10.1128/spectrum.03385-22) devised a neat approach for simultaneous introduction of multiple mutations in multiple fragments. Perhaps authors should consider removing or rephrasing this statement.

We have rephrased the statement in the introduction to indicate that the CPER method is fast when a template with desired mutations is available, but requires cloning of individual fragments into plasmids for a large number of mutations (page 3, line 88-93). In addition, we have cited the indicated publication to demonstrate that there are new applications of CPER that may overcome this limitation.